# Summary of the Transformational Relationship between Point Load Strength Index and Uniaxial Compressive Strength of Rocks

**Meiqian Wang [1], Wei Xu [1], Dakun Chen [2], Jianguo Li [3], Hongyuan Mu [3], Jian Mi [3] and Yonghong Wu [1,*]**

1   Faculty of Civil and Architectural Engineering, Kunming University of Science and Technology, Kunming 650500, China
2   Yunnan Guangsha Planning Architectural Design Institute Company, Chuxiong 675000, China
3   Yunnan Institute of Water & Hydropower Engineering Investigation, Design and Research, Kunming 650021, China
*   Correspondence: 20130112@kust.edu.cn

**Abstract:** The point load test is an effective and rapid way to predict rock strength. Regarding the investigation of point load strength and the failure characteristics of rock, the point load test's advantages and application scopes are introduced in this paper. According to the three main components—the rock itself, the size effect, and the loading cross-sectional area—the point load strength's influencing factors and mechanisms on rock failure were analyzed, followed by expounding the significant effect of the technology of the point load test on evaluating engineering safety and stability. Based on previous scholars' research results, there is a strong correlation between the point load strength and the uniaxial compressive strength. The parameters of the rocks from different regions and different sediments were summarized via substantial field and indoor testing. The functional relationship (mainly including the linear function, quadratic function, exponential function, power function, and logarithmic function) between the point load strength and the uniaxial compressive strength was obtained by mathematical statistical analysis. Finally, the challenges regarding the point load test were discussed, and accordingly, suggestions for future research were provided.

**Keywords:** point load test (PLT); size effect; point load strength (PLS); uniaxial compressive strength (UCS)





## 1. Introduction

The uniaxial compressive strength (UCS) of rock has been recognized as one of the major technical factors of rock-related engineering constructions [1–3], and it has also been regarded as the basic parameter of geotechnical engineering designs and constructions [4]. Moreover, UCS has been used in various aspects of the rock-mass classification systems and rock engineering design [5–7]. The correctness of its experimental data exerts a significant effect on engineering safety and expenditure [8,9]. The direct and indirect methods were more common for determining the UCS of rocks. The direct measurement of UCS is both time-consuming and expensive [10], while the indirect method is cheaper, faster, and more convenient to perform, both the laboratory and in the field [11]. The direct method aims at determining the compressive strength, tensile strength, shear (fracture) strength, rock failure/fracture mechanisms, and strength criteria. The determination of uniaxial compressive strength requires precise testing devices and high-quality rock-core samples [12], but these rock parameters could not always be directly obtained through the traditional coring method [13]. Therefore, a series of indirect testing methods have also been employed to estimate the UCS of rocks [14–16], such as the PLT, Schmidt hammer test, block punch index test, Equotip hardness test, needle penetration test, and acoustic frequencies analysis [17–23]. The PLT is one of the indirect methods for estimating the

UCS of rocks [24]. The point load strength (PLS) test has been widely used in global rock engineering because of its easy preparation, low cost, and simple operation [25].

The instrument for measuring PLS consists of portable testing equipment, which is applicable to all kinds of rocks. It is not necessary to cut or grind the testing samples in the field or laboratory [26]. Moreover, these rock samples can be taken from drilling and coring sites, outcrops, exploration pits, adits, roadways, or other caverns [27]. The point load instrument is small in size and convenient to carry, and can also be used to measure cylindrical rock cores and irregularly shaped samples [19]. In addition, the rock samples do not need to be pre-processed, making it possible to measure the strength of weak and broken rocks.

## 2. Point Load Calculation Method

### 2.1. Influencing Factor of Rock Point Load Failure

The PLT method involves placing the rock sample between the upper and lower conical pressing plates with the ball ends, applying the concentrated load on the sample until it is destroyed, then obtaining both the point load strength index ($I_{s(50)}$) and the strength anisotropy index of the rock, of which the entire process is regarded as an index test for the strength classification of rock materials [28]. The failure of rock under the point loading is a gradual process, and its initiation occurs at two loading points, producing symmetrical, up and down, local compressive stress. With continuous loading, the compressive stress constantly decreases, while the upper-lower symmetrical tensile stress is generated near the loading point. The compressive stress and tensile stress keep approaching the center of the loading axis until the two areas coincide with each other. As the tensile strength of the sample is far lower than the compressive strength, the sample is mainly damaged under the force of tensile stress, along with the specific influence of compressive stress [29,30].

The failure state of rock under point load [31,32] could be divided into four modes, i.e., the single-sided failure, triple connection failure, twist failure, and single-sided (inclined) failure [33]. The rock sample produces a vertical tensile crack under a low constraint pressure, while it generates the bending and torsional deformation failure under a high constraint pressure [34]. There are several factors affecting the changes in its strength value, mainly including:

① The influence of the rock itself

For the collected rock patterns, we often encounter the influence of layered structure, joint fissure development, and other rock characteristics on the test strength; the particle size and mineral composition of the rock also have a significant influence on the test results. Xu et al. [35] concluded that there were apparent joint cracks along the failure surface of the rock lump damaged by joints; the rock lump with straight damage shows elastic failure, and its failure surface is both flat and fresh; the PLS of a rock lump with bending failure is larger, and its failure surface shows tiny joints.

② The effect of sample shape

The size of the sample also exerts a significant effect on the PLS. Zhu et al. [36] studied the variation patterns in the shape coefficient, loading point spacing, and PLS; the research of Hawkins [37,38] showed that the transformation coefficient was affected by the shape, size, and water content of the sample; the studies of Wong [39] and Koohmishi [40] also found that the PLS was affected by both size and shape; Yao et al. [41] conducted a series of point load tests on the samples of red bed siltstone with different shapes (cylinder, square, and irregular), and revealed the characteristics of high dispersion of the point load coefficient.

③ The effect of sample size

The factors affecting the size of point load samples are mainly reflected in the aspects of height-diameter ratio D/L (as shown in Figure 1) and loading point spacing D. Substantial tests have proved that when the D/L of rock samples is beyond 0.5, its $I_{s(50)}$ will basically remain a constant. If the D/L is below 0.5, its PLS will decline with the decrease in

this height-diameter ratio. Generally speaking, the greater the distance of the point load imposed on rocks, the greater the strength value of the rock samples. However, for some rocks with small strength, the change range of the $I_{s(50)}$ is not obvious with the increase in their loading point spacing [42]. Koohmishi [43] identified the relationship between the PLS index and the equivalent core diameter, as listed in Table 1. As an observation, the size effect also exerts a significant influence on the PLS [2], which decreases with the increase in sample size [44].

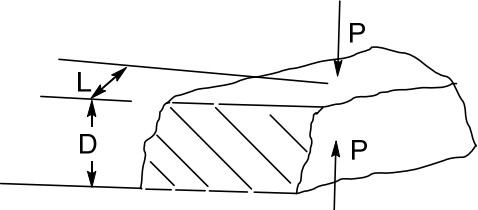

**Figure 1.** Height-diameter ratio.

**Table 1.** Relationship between load strength index and equivalent core diameter.

| Serial No. | PLS Index Formula | Correlation Coefficient ($R^2$) | Rock Type |
|------------|-------------------|-------------------------------|-----------|
| 1 | $I_s = -0.516De + 31.423$ | 0.897 | Basalt |
| 2 | $I_s = -0.824De + 50.364$ | 0.835 | Dolomite |
| 3 | $I_s = -0.137De + 11.776$ | 0.764 | Limestone |
| 4 | $I_s = -0.105De + 9.209$ | 0.782 | Marl |
| 5 | $I_s = -0.414De + 25.710$ | 0.828 | Quartzite |
| 6 | $I_s = -0.606De + 38.694$ | 0.892 | Lava |

④ The effect of sample's failure load area

The failure load area refers to the cross-sectional area of failure formed by the rock fracture surface during the loading process. For the regular rock samples, the failure cross-sectional area is D × D. For the irregular rock samples, the area is equivalent to the area of a square with a side length (diameter) of De [45]. The failure load area of a regular rock sample exerts little effect on its PLS, while the failure load area of an irregular rock sample has a certain effect on its PLS. By studying the failure load area of a rock sample, the additional influences of size and shape can be reduced, to a certain extent.

### 2.2. Research on Failure Mechanism Using the Point Load Test

To study the failure mechanism of the sample under point loading, it is necessary to understand the initially stress distribution within the sample under point loading. Brook [46] and Gercek [47] have conducted many studies with this objective, and their conclusions are basically the same principle. Gong et al. [48] obtained the linear energy storage and energy consumption law in the process of rock tensile failure through the loading and unloading tests of red sandstone. Reichmuth [49] studied the point load failure mechanism of rock samples (as shown in Figure 2) and pointed out that: under point loading, the central region of the rock would form a certain range of tensile stress regions and compressive stress regions; as the load increased continuously, the rock sample would be damaged gradually. Near the loading cone, cracks were formed under the action of compressive stress. Numerous cracks showed coalescence with each other, generating a sliding line. As the crack sliding line gradually deepened, the sample eventually broke under the action of tensile stress.

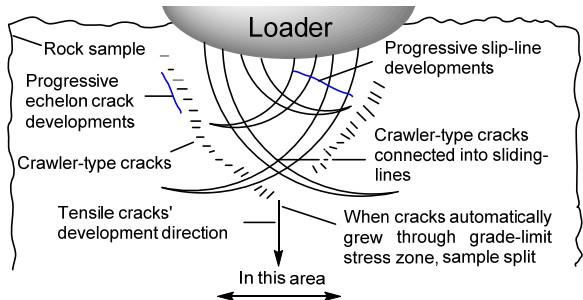

**Figure 2.** Failure mechanism of point load.

Using the finite element method, Peng [50] determined that there was a wide range of tensile stress within the rock sample, which was a symmetrical biaxial tensile stress that would cause the fracture of the rock sample. Masoumi et al. [44] studied rock types of different geological origins (including sedimentary, igneous, and metamorphic), and concluded that the rock failure was changed from pure tensile fracture to a combination of shear and tensile fractures. Xiao et al. [47] studied the influence of temperature on sandstone failure. The basis of the PLT is to uniaxially break the rock sample and determine the corresponding stress in its fracture process. The test procedure is interpreted by ISRM as the diametral test, axial test, block test, and irregular lump test (Figure 3).

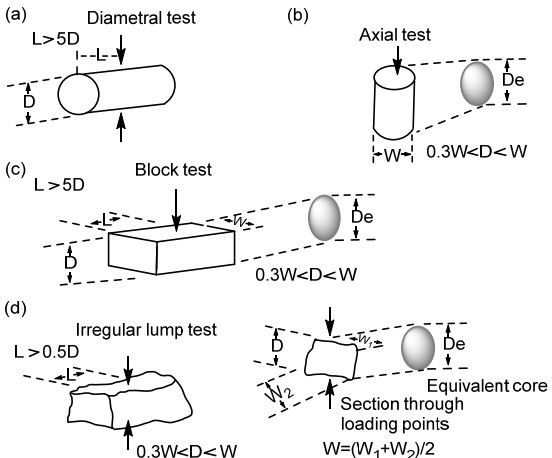

**Figure 3.** Sample dimensions: (**a**) diametral, (**b**) axial, (**c**) block, (**d**) lump [30].

*2.3. Research on Calculation Method of Point Loads*

2.3.1. Calculation Formula

In 1972, Broch and Franklin [28] conducted the failure model analysis of a cylinder sample along the diameter direction and obtained the PLS formula:

$$I_s = F/D^2 \tag{1}$$

where: $I_s$ is the PLS, F is the failure load, and D is the sample diameter.

In 1975, Broch and Franklin also proposed the concept of core equivalent diameter De and modified the above PLS formula as follows:

$$I_s = F/De^2 \tag{2}$$

where: De is the equivalent diameter of the sample.

In 1980, Kaharaman [51] first proposed the adoption of the method of equivalent diameter for the calculation formula of the point load of irregular rocks.

In 1985, ISRM formally proposed a newly recommended method for measuring the PLS, using the equivalent diameter to correct the $I_{s(50)}$ of the standard 50 mm rock sample at the longitudinal test point [30], by means of the correction factor T to obtain the following formula:

$$T = (De/50)^{0.45} \tag{3}$$

$$I_{s(50)} = T \cdot I_s \tag{4}$$

In 1986, Xiang et al. [42] pointed out that the previous formula of PLS did not consider the influencing factor of the failure surface's size and suggested the alternative formula as follows:

$$I_s = P/A_f \tag{5}$$

where, $A_f$ is the failure surface area, and P is the failure load.

In 1987, Li et al. [52] proposed a method for calculating the PLS of irregular lump samples and considered that the volume index could be applied to the calculation by the following formulas:

$$I_s = P/(V^{2/3}) \tag{6}$$

$$I_s = P/(V^{2/3} \cdot A_f^{1/2}) \tag{7}$$

where, V is the volume.

For the process of transforming $I_s$ into $I_{s(50)}$, it is practicable to apply the D-lg$I_s$ curve method, the dimension–correction curve proposed by Broch and Franklin, and the Hassani formula [53]:

$$I_s = P/(D^2)$$
$$\lg I_{s(50)} = 0.256 + \lg I_s - 1.008 \cdot e^{-0.0274 \cdot D} \tag{8}$$

where: $I_s$ is the uncorrected index of PLS; P is the failure load; D is the distance between different loading points; $I_{s(50)}$ is the standard index of PLS.

In China, scholars mainly calculate $I_{s(50)}$ using the following formulas [27]:

$$I_s = P/De^2 \tag{9}$$

$$De^2 = 4WD/\pi \tag{10}$$

$$I_{s50} = I_s \cdot F \tag{11}$$

$$F = (De/50)^{0.45} \tag{12}$$

where: De is the equivalent diameter; W is the width of the minimum section passing through the two loading points; F is the correction coefficient.

2.3.2. Research Status of Correction Index

As for the correction index m, the ISRM [30,54] proposed that the slope (n) of the $\log(P)$-$\log(De^2)$ relationship could be used to determine the **m** value of the size correction factor as $m = 2(1 - n)$. Therefore, it can be calculated under the vertical bedding that $n = 0.7793$; $m = 0.44$; under parallel bedding, $n = 0.79$; $m = 0.42$ [27]. Wong et al. [39] studied granite samples with different weathering degrees, finding that the actual correction index m obtained by the regression of the strength data of samples with different sizes was quite different from its recommended value in the specification. Yin et al. [55] also found that in the size correction function, the correction index value m of slightly-weathered granite was around 0.443–0.600, and that of moderately-weathered granite was between 0.545–5.562. Li et al. [56] used different loading methods (axial test and diametral test) to carry out the PLT and obtained a correction index of 0.5. Yao et al. [57] conducted both the PLS test and UCS test on rock samples with vertical and parallel beddings, respectively, revealing that the correction index of gneiss was $m = 0.44$ under vertical bedding and $m = 0.42$ under parallel bedding. Dai et al. [58] carried out the axial tests on three disc-shaped samples

with different coring diameters and obtained the results that the correction indexes m of marbles and red sandstones were 0.44 and 0.53, respectively.

Therefore, it is not difficult to determine that there is a large gap between the correction indexes obtained by different scholars, which did not share a sound universality due to their regional characteristics.

### 3. Study on Transformation between PLT and UCS

The PLT is a test method—which is easy to operate and prepare—for the rapid prediction of the UCS of rocks. The PLT is widely used, with many corresponding research results. Protodyakonov [59] first put forward the idea of PLT with irregular blocks, then D'Andrea [29] and Franklin [30] studied the transformation between the PLS and UCS of rocks. Broch et al. [28] conducted point load tests on standard samples with different heights H and diameters D and found that H/D is constant; when both H and D increased, the PLS basically remained unchanged, but when D remained unchanged, the PLS decreased significantly with the increase in H/D. Gunsallus [60] proved that the PLS shared certain correlations with the strength values of other tests for the fracture toughness test, UCS test, PLT and Brazilian tensile test. Peng [50] studied the stress state of the sample based on the finite element method, finding that when H/D < 1, the internal stress state of the sample could remain stable.

In 1985, on the basis of previous research results, Franklin [30] revised the PLT specifications, which have been widely recognized and applied by the majority of scholars. Since then, many of them have studied the PLT from different angles, perspectives, and methods, including the transformation between PLS and UCS [39,61], size correction, and shape correction [40,62,63]. These studies showed that the PLS index shared a good correlation with the UCS [64–67]. Therefore, the PLT method can also reflect the strength characteristics of rocks. Li et al. [54] discussed the stress distribution laws of five typical irregular-shaped rock blocks by using the PLT method.

Moreover, Kahraman [68] tried to verify the correlations between the UCS value and the differing results of the PLT, rebound test, acoustic test, and impact strength test through the correlation analysis between the UCS value and other test values using the least square method, which concluded that the $I_{s(50)}$ of coal had a strong linear relationship with the UCS value, and the results of the rebound test and the acoustic wave test shared the nonlinear correlations with the compressive strength of the rocks. Cobanoglu et al. [69] studied the relationships of UCS with the PLS index, P-wave velocity, and Schmidt hardness, and explored the influence of core diameter on the PLS index by testing five core samples of sandstone, limestone, and cement mortar with different diameters. Kohno et al. [70] conducted the point load tests on eroded rock lumps on the surface and on coring of the same kind of rocks, indicating that PLT was an effective method to test the rock strength, and that the $I_{s(50)}$ was a reliable index to measure the rock strength. Kahraman [49] implemented the point load tests on soft rocks such as the pyroclastic rocks, and studied the relationship between their UCS of less than 50 MPa and thier $I_{s(50)}$, finding that the exponential transformation formula was more consistent with the transformation of these two indexes of pyroclastic rocks. Azimian et al. [16] aimed to describe the relationship between UCS and $I_{s(50)}$ of marl, by conducting the regression analysis on P-wave velocity of marl and established an empirical formula. Karaman et al. [64] determined the applicability of $I_{s(50)}$ and the Schmidt hammer hardness value to the UCS and tensile strength of rocks, also using regression analysis. Kaya et al. [5] studied the transformation between UCS and $I_{s(50)}$ of three types of rocks, with the strength transformation factor k obtained by the zero-intercept regression analysis, formula, and graphical methods. Oztur et al. [71] carried out the indoor uniaxial compressive tests and field point load tests on the natural alkalis and interlayers of volcanic sedimentary rocks. Nagappan [65] conducted a field simulation on weak rock mass to study the correlation between UCS of the jointed rock mass and the point load index. Yin et al. [55] implemented point load tests on 754 granites

and established the correlation between the $I_{s(50)}$ and UCS of irregular lump tests through the appropriate size correction functions.

Furthermore, Zhu et al. [36] studied the change laws of shape coefficient, loading point spacing, and PLS by carrying out the PLT on soft phyllite, discussing the correction method for calculating the PLS of irregular soft phyllite based on its minimum cross-sectional area. Luo et al. [72] also analyzed soft phyllite samples with different weathering degrees based on the PLT and found that after the soft phyllite was soaked, the water absorption and saturated water absorption of the rock lump gradually decreased with the increase in saturated UCS, while the water saturation coefficient increased slightly. The disintegration resistance index decreased gradually with the increase in dry-wet cycles, and the lower the rock strength, the faster the decrease. Feng et al. [73] took gneiss as the research object and carried out the wave velocity test, UCS test, and PLT on the rock samples with vertical bedding and parallel bedding, which indicated that both the wave velocity and the strength of gneiss showed anisotropic characteristics in the cases of vertical bedding and parallel bedding. Sha et al. [74] adopted the method of equivalent diameter to effectively reduce the error in traditional calculations, thus improving the accuracy of measuring the PLS. Zheng et al. [75] used the methods of statistics and data fitting to analyze the test data of basalts in the Mumbai Peninsula, revealing the functional relationships among the rock density, UCS, PLS, uniaxial tensile strength, elastic modulus, and Poisson's ratio of basalts. Zhou et al. [76] studied the functional relationships between the PLS and UCS of Cenozoic red sandstone through UCS testing and PLT. Li et al. [77] identified the numerical relationship between the PLS and the UCS by coupling the dispersed mesh elastic model and the discontinuous deformation analysis elastic model, which achieved the automatic calibration of these parameters by using the improved Newton method, obtaining the UCS of the rock. Koohmishi [39] conducted the point load tests on basalt, dolomite, limestone, marl, and volcanic rocks, obtaining the relationship between their $I_{s(50)}$ and their equivalent core diameter. Shahla et al. [78] took tuff samples from the Atashkuh quarry in Mahallat (central Iran) to study the effect of PH on the physical and mechanical properties of tuff, including its porosity, point load index, and tensile strength, finding that with the increasing number of dry-wet cycles of the $H_2SO_4$ solution, the porosity increased, while the point load index and tensile strength both decreased by different degrees. Goulet [79] revealed the effect of core alteration on the anisotropy of rock mass strength using point load tests. Mehdi [80] also studied the effect of reinforced soils on the mechanical properties of cohesive soils through point load tests. Palchik et al. [81] sampled porous rocks for mechanical testing to study the effect of porosity on PLS and summarized the relationship among these mechanical strengths, with the obtained results being consistent with the PLS measurement method recommended by the ISRM.

For the relationship between PLS and UCS, researchers all believe that there is an obvious empirical correlation between the $I_{s(50)}$ and the UCS. They have carried out substantial studies using field tests, summarized the rocks in different regions, and established a variety of transformational relationships, including the zero intercept linear function, non-zero intercept linear function, exponential function, power function, quadratic function, and logarithmic function models. Their test outcomes are different from each other, and the correlations are weak. Their final research results are mainly applicable for one or several types of rocks in a certain country or region, implying that these transformation formulas all have regional limitations. The empirical transformation formulas using PLS to predict UCS are listed, according to the research results of different scholars using different rock samples.

### 3.1. Linear Function Relationship

Researchers from different countries and regions carried out the point load tests according to the different petrogenesis of the rocks, including different sizes and shapes. It is concluded that the $I_{s(50)}$ and UCS share a relationship of zero intercept linear function [2,8,10,14,17–19,28,30,37,38,41,42,55–58,61,70,71,81–100] on the basis of their research on basalt, diabase, granite, tuff, mudstone, sandstone, limestone, diorite, andesite, calcare-

ous sandstone, sandstone, dolomite, marble, gneiss, and red sandstone, UCS = K·$I_{s(50)}$. Among them, the coefficient K is between 7 and 68, while the coefficients obtained by most researchers is concentrated between 15 and 25, which is close to the coefficient recommended by the ISRM [50] of between 20 and 25. However, the results of many researchers and the similar studies demonstrated that the coefficients presented a broader range [19].

Numerous scholars have also studied basalt, granite, quartzite, schist, siltstone, fine sandstone, medium sandstone, coarse sandstone, sandy mudstone, mudstone, pyroclastic rock, shale, sandstone, limestone, conglomerate, marl, gneiss, diabase, diorite, limestone, calcareous sandstone, and other rocks, before concluding that the relationship between $I_{s(50)}$ and UCS is a non-zero intercept linear function [11,14,24,29,60,68,91,94–96,101–112], UCS = β·$I_{s(50)}$ + α. Where coefficient β is between 3.49 and 27.42, and α is between −39.64 and 89.87. The coefficients of different rocks obtained by different researchers all exhibited large dispersion and poor representativeness.

However, whether the conversion relationships between $I_{s(50)}$ and UCS were obtained from different rock types via zero intercept linear functions or non-zero intercept linear functions, they are obvious different and not representative.

### 3.2. Other Functional Relationships

Some researchers also based their studies on the sedimentary type of rocks to conduct the point load tests and UCS tests, with different water content and loading conditions, before obtaining the other transformational relationships between PLS and UCS, as listed in Table 2.

**Table 2.** Other functional relationships.

| Functional Relationship Type | Author | Functional Relationship | Major Rock Type | Remarks |
|---|---|---|---|---|
| Quadratic function | Tugrul et al. [88] | UCS = 3.86$(I_{s(50)})^2$ + 5.65$I_{s(50)}$ | Magmatic rock | — |
| | Quane et al. [89] | UCS = 3.86$(I_{s(50)})^2$ + 5.56$I_{s(50)}$ | — | — |
| | Sha Peng et al. [74] | UCS = 0.31$(I_{s(50)})^2$ + 7.01$I_{s(50)}$ (R = 0.966) | Tuff | — |
| | | UCS = −0.94$(I_{s(50)})^2$ + 24.56$I_{s(50)}$ (R = 0.966) | Diorite | — |
| | Sha P. et al. [76] | UCS = 0.14$(I_{s(50)})^2$ + 13.25$I_{s(50)}$ (R = 0.3) | Igneous rock | — |
| Exponential function | Diamanti et al. [14] | UCS = 16.45exp(0.39$I_{s(50)}$) | — | — |
| | Sheraz et al. [11] | UCS = 85.52exp(0.718$I_s$) (R = 67%) | Dolomite | — |
| | Kahraman [24] | UCS = 1.99exp(1.18$I_{s(50)}$) (R = 0.92) | Pyroclastic rock | Saturated |
| | | UCS = 2.27exp(1.04$I_{s(50)}$) (R = 0.93) | | Natural |
| | | UCS = 2.68exp(0.93$I_{s(50)}$) (R = 0.93) | | Dry |
| | Li Shao Qian et al. [113] | UCS = 6.46exp(0.56$I_{s(50)}$) | Limestone | Axial |
| | | UCS = 9.80exp(0.47$I_{s(50)}$) | | Diametral |
| Power function | Tsiambaos et al. [90] | UCS = 7.3$(I_{s(50)})^{1.71}$ ($R^2$ = 0.906) | — | — |
| | Santi et al. [114] | UCS = 12.25$(I_{s(50)})^{1.50}$ ($R^2$ = 0.985) | — | — |
| | Diamanti et al. [14] | UCS = 17.81$(I_{s(50)})^{1.06}$ ($R^2$ = 0.906) | — | — |
| | ASTM [115] | UCS = 17.81$I_{s(50)}^{1.06}$ | Serpentinite | — |
| | Ministry of Water Resources of the People's Republic of China [27] | UCS = 22.82$(I_{s(50)})^{0.75}$ ($R^2$ = 0.90) | — | — |
| | Sheraz et al. [11] | UCS = 202.71$I_s^{0.633}$ (R = 80%) | Dolomite | — |
| | Kahraman [24] | UCS = 7.73$(I_{s(50)})^{1.25}$ (R = 0.910) | Pyroclastic rock | Dry |
| | | UCS = 8.61$(I_{s(50)})^{0.95}$ (R = 0.910) | | Saturated |
| | | UCS = 8.66$(I_{s(50)})^{1.03}$ (R = 0.922) | | Natural |
| | Chen Jiaqi et al. [116] | UCS = 22.72$I_{s(50)}^{0.82}$ ($R^2$ = 0.860) | Sandstone, mudstone, and limestone | Irregular |
| | | UCS = 26.24$I_{s(50)}^{0.72}$ ($R^2$ = 0.860) | | Regular |

**Table 2.** *Cont.*

| Functional Relationship Type | Author | Functional Relationship | Major Rock Type | Remarks |
|---|---|---|---|---|
| Logarithmic function | Grasso et al. [103] | $UCS = 100\ln I_{s(50)} + 13.9$ | — | — |
| | | $UCS = 17.04\ln I_{s(50)} + 9.29$ | — | — |
| | Kilic et al. [107] | $UCS = 100\ln(I_{s(50)}) + 13.9$ (R = 0.990) | — | — |
| | Kahraman [24] | $UCS = 17.04\ln(I_{s(50)}) + 9.29$ (R = 0.750) | Pyroclastic rock | Dry |
| | | $UCS = 7.27\ln(I_{s(50)}) + 11.7$ (R = 0.750) | | Saturated |
| | | $UCS = 10.28\ln(I_{s(50)}) + 12.32$ (R = 0.730) | | Natural |

Comparing the results of these researchers, the functional transformations between the point load tests and the UCS are mainly linear functions, while the transformation types of quadratic function, exponential function, power function, and logarithmic function are relatively fewer, but the data results of each functional transformation are all very dispersed. These different functional relationships between PLS and UCS have different shortcomings, whereas the field conditions are not simulated during all the tests. The PLT can be used for both field testing and indoor testing, while the UCS test is only applicable for indoor testing. During the transportation and processing of rock samples, the grains [70] within the rock mass may be damaged; consequently, the test data obtained from rock blocks with joints [67,111,117] and fissures [118] might become more dispersed.

## 4. Discussions

In addition to the influences of rock properties [40,119,120], size effect [43,44], loading point spacing [25], loading cross-sectional area [99], and rock loss [121], the strength of rock is also affected by factors such as alteration [81], porosity [122], water content [66,70,71,123,124], temperature [17,125], the freeze-thaw cycle [126], the loading cycle [127], and the dry-wet cycle [80]. As a result, the transformations between $I_{s(50)}$ and UCS obtained by researchers in different countries are quite distinct. Similarly, the long-term strength of rocks in a natural state can be estimated by the PLT method [128]. In the previous studies, only the influence of a single factor on rock strength has been considered, there are few studies on rock strength coupling multi-factor conditions, and artificial neural network analysis, multivariate models, Bayesian analysis, fuzzy theory analysis, and other methods for studying the conversion relationship between $I_{s(50)}$ and the UCS have not been reported.

In the engineering of slopes, foundation pits, and tunnels, the role of water cannot be ignored, and most of the geological engineering environments are very complex, involving water–rock coupling, high ground stress, etc.; thus, the research on PLS faces severe challenges. In the natural state, most rock stress is triphasic [129], but during the tests on these rocks, the influence of their confining pressure was ignored. Therefore, it is also recommended to adopt the true triaxial experiment for the study of UCS testing, and the PLT should simulate the three-dimensional stress conditions, which also produces severe challenges regarding the entire measurement.

## 5. Conclusions

Based on previous studies, this paper summarizes the developments and achievements of PLTs. The conclusions are as follows:

1. The failure of rock under point load is a gradual process. The failure starts at the upper and lower loading points, producing symmetrical local compressive stress. With the continuous loading, the compressive stress constantly decreases.

2. Research is carried out regarding the fundamental problems of point load tests. Based on the analysis of the research progress, the calculation formulas and influencing factors of PLS of the rock are summarized, including the influences of rock size, shape,

and compression cross-sectional areas on PLS. Meanwhile, the influences of rock properties and environments on PLT results were also discussed.

3. The functional relationships between $I_{s(50)}$ and UCS were established according to the research results of previous researchers on different types of rocks by using mathematical statistics methods. However, to a certain extent, these functions are all relatively dispersed, when coupled with regional and other factors; therefore, the obtained functional relationships are of poor general applicability.

The reported conversion relationships between $I_{s(50)}$ and UCS are not representative; however, more precise and representative methods have not yet been reported.

**Funding:** This research was funded by [The Major Science and Technology Special Plan of Yunnan Province Science and Technology Department] grant number [202002AF080003], [The sponsorship from National Natural Science Foundation of China] grant number [51669009] and [The Yunnan Institute of Water and Hydropower Engineering Investigation, Design and Research] grant number [KKKF0202006249].

**Institutional Review Board Statement:** Ethical review and approval were not obtained for this study because no intervention was applied to the participants.

**Informed Consent Statement:** Informed consent was obtained from all subjects involved in the study.

**Data Availability Statement:** The data have been collected using public funds, but at the time of the publication of this paper, they have not been shared in a repository.

**Acknowledgments:** This received support from the Major Science and Technology Special Plan of Yunnan Province Science and Technology Department (202002AF080003). The authors are grateful for the sponsorship of the National Natural Science Foundation of China (51669009) and give thanks to the Yunnan Institute of Water and Hydropower Engineering Investigation, Design, and Research (KKKF0202006249).

**Conflicts of Interest:** The authors declare no conflict of interest.

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
