# Peer review of "Summary of the Transformational Relationship between Point Load Strength Index and Uniaxial Compressive Strength of Rocks"

_sustainability, doi:10.3390/su141912456_

Round 1

Reviewer 1 Report

Dear Authors

The paper entitled “Summary of transformation between point load strength and uniaxial compressive strength of rocks” accurately considered for reviewing. The topic seems a review paper. As a total decision the topic is interesting and could be published as original research paper after following major revisions. All comments should be corrected accurately:

1.      The writing and grammatical factors of paper is acceptable but could be improved for enhancing the readability of paper.

2.      The quality of Figures 1, 2, and 3 are not acceptable. All figures must be changed absolutely.

3.      The format of tables should improve.

4.      What is the main goal of authors for selecting this issue to research?

5.      Is there any tests are applied by authors for checking various types of correlation between PLT and UCS? Or just previous research reviewed?

6.      References citation have many mistakes in text. All of them should be corrected.

7.      Literature Review have not covered other methods for predicting UCS. Acoustic frequencies analysis is one of accurate methods by this purpose. These methods should be covered in introduction part citing following papers:

a.      Investigating an innovative model for dimensional sedimentary rocks characterization using acoustic frequencies analysis during drilling; M Yari, R Bagherpour; Rudarsko-geološko-naftni zbornik 33 (2), 25-25

b.      Kumar BR, Vardhan H, Govindaraj M (2010) Estimating rock properties using sound level during drilling: field investigation. Int J Min Mineral Eng 2:169–184

c.      Implementing acoustic frequency analysis for development the novel model of determining geomechanical features of igneous rocks using rotary drilling device; M Yari, R Bagherpour; Geotechnical and Geological Engineering 36 (3), 1805-1816

d.      Developing a novel model for predicting geomechanical features of carbonate rocks based on acoustic frequency processing during drilling; M Yari, R Bagherpour, M Khoshouei; Bulletin of Engineering Geology and the Environment 78 (3), 1747-1759

8.      References are checked. Reference list is acceptable and covers old and new papers.

Reviewer 2 Report

The paper concerns a revision of point load test's advantages and application scope in rocks mechanic. Although emphasis is placed on the correlation between the point load strength and the uniaxial compressive strength, it would has expected, considering the nature of the dynamics of the field of study, the multivariable relationship and its modeling, in order to study both the individual and multiple impact of the relationship between PLT and UCS. In addition, although statistical models and univariate regressions are helpful explaining the relation between two variables, it could be of interest for readers introduce additional tools used by another authors for model this relations, tools like machine learning algorithms.

Others form suggestions, were presented below.

·        Rewrite equations below line 181, potential typing errors.

·        In line 207, Decide between using the name of the variable or its acronym, both are redundant, or failing that, consider one of them in parentheses. Idem line 234. Correct this throughout the document.

·        First paragraph of section 3 too long, this makes it difficult to follow the reading. You should divide it into at least 2 or 3 paragraphs.

·        Check cohesion of line 287

·        In Section 3.1, in each paragraph you must assign the specific reference to each mineral (the list shown in lines 290 and 298-299 is confused for readers).

·        Check grammar of line 303, it is confusing

·        Line 309-310… What did you mean with "the results of each function are very discret"?

·        Discussion section is somewhat poor. I propose to complement it with advances or novel approaches applied to the study of the process in question, such as multivariate models, or machine learning applications.

·        The conclusions provided by the authors seem more like final ideas of the introductory section. The conclusions must be related to the objectives of the study, I cannot see this relationship in its conclusions (rather, certain obvious things are declared). Their conclusions must be complemented.

Additional general suggestions:

·        Improve quality of all figures.

·        English language usage is poor and should be improved throughout the manuscript regarding grammar and orthography.

Round 2

Reviewer 1 Report

Dear editor

The paper in present form could be accepted as an original paper. All mentioned comments in first round of review are edited. 

I am so sorry for late answer. 

Best wishes